# Prevalence of Sarcopenia in Women with Breast Cancer

**DOI:** 10.3390/nu14091839

**Published:** 2022-04-28

**Authors:** Delia Morlino, Maurizio Marra, Iolanda Cioffi, Lidia Santarpia, Pietro De Placido, Mario Giuliano, Carmine De Angelis, Simone Carrano, Annarita Verrazzo, Giuseppe Buono, Marianna Naccarato, Olivia Di Vincenzo, Enza Speranza, Sabino De Placido, Grazia Arpino, Fabrizio Pasanisi

**Affiliations:** 1Internal Medicine and Clinical Nutrition Unit, Department of Clinical Medicine and Surgery, Federico II University Hospital, 80131 Naples, Italy; delia.morlino@unina.it (D.M.); iolanda.cioffi@unina.it (I.C.); lidia.santarpia@unina.it (L.S.); marianna.naccarato@unina.it (M.N.); olivia.divincenzo@unina.it (O.D.V.); enza.speranza@unina.it (E.S.); pasanisi@unina.it (F.P.); 2Oncology Unit, Department of Clinical Medicine and Surgery, Federico II University Hospital, 80131 Naples, Italy; pietrodep91@gmail.com (P.D.P.); m.giuliano@unina.it (M.G.); carmine.deangelis1@unina.it (C.D.A.); sim1.carrano@gmail.com (S.C.); annarita.verrazzo@virgilio.it (A.V.); giuseppe.buono88@gmail.com (G.B.); deplacid@unina.it (S.D.P.); grazia.arpino@unina.it (G.A.)

**Keywords:** breast cancer, sarcopenia, body composition, bioimpedance analysis (BIA), hand grip strength (HGS), phase angle (PhA)

## Abstract

Sarcopenia is a common finding in patients with cancer and potentially influences the patient’s outcome. The aim of this study was to evaluate the prevalence of sarcopenia, according to the European Working Group on Sarcopenia in Older People, in a sample of women with breast cancer (BC) and a BMI lower than 30 kg/m^2^. This cross-sectional study was conducted in patients with BC, stage 0-III, and receiving therapy for BC; the women were recruited at the Department of Clinical Medicine and Surgery, Federico II University, Naples, Italy. A control group with similar age and BMI was selected from the internal database. Anthropometry, bioimpedance analysis (BIA) and hand grip strength (HGS) were measured to detect sarcopenia. A total of 122 patients (mean age 49.3 ± 11.0 years, BMI 24.6 ± 3.0 kg/m^2^) and 80 healthy controls were analyzed. Sarcopenia was found in 13.9% patients with BC, while none of the subjects in the control group was sarcopenic. By comparing BC patients with and without sarcopenia and the control group, the fat-free mass of sarcopenic BC patients were significantly lower than those of both non-sarcopenic BC patients and the control (*p* < 0.05). The phase angle was also significantly lower in sarcopenic patients (−0.5 degrees, *p* = 0.048) than in the control group. Considering the prevalence of sarcopenia in patients with BC, our findings suggest the usefulness of body composition and HGS evaluation for early screening of sarcopenia to reduce the risk of associated complications.

## 1. Introduction

Breast cancer (BC) is the most frequently diagnosed carcinoma and the second leading cause of cancer death among females worldwide [1]. According to a recent publication by the American Cancer Society, BC incidence rate has been increasing by 0.3% per year, but its death rate continues to decline, falling 40% over the last thirty years [2]. According to the Italian Society of Medical Oncology, BC incidence has increased slightly, also in Italy; in 2020, approximately 55,000 new cases of BC were recorded (+0.3% per year), whereas mortality decreased (−0.8%) thanks to the improved screening programs and high-quality multidisciplinary treatment of the disease [3]. In the last decade, body composition measurements have gained greater attention in oncology research, since sarcopenia is a common finding in patients with cancer, potentially influencing chemotherapy toxicity and the patient outcome [4]. Different parameters can be evaluated to assess body composition by bioimpedance analysis (BIA), such as fat-free mass (FFM), fat mass (FM) and phase angle (PhA). The latter is a raw BIA parameter, inversely correlated with disease severity, inflammation and malnutrition in several clinical conditions, as well as in cancer patients. PhA is claimed to be a good nutritional marker, especially for cancer patients [5], and also represents an indicator of sarcopenia [6].

Sarcopenia is a syndrome characterized by the progressive reduction in both muscle mass and muscle strength, which is accompanied by a reduction in the quality of life and an increased risk of adverse outcomes [7]. Despite being typical of the elderly population, sarcopenia also occurs earlier in life due to the presence of acute or chronic diseases [8,9].

In 2010, the European consensus [10] distinguished primary sarcopenia from secondary sarcopenia. The first condition occurs when no other clear etiologies are found, and advanced aging is the main cause, whereas secondary sarcopenia is associated with some chronic clinical conditions (inflammatory and chronic diseases, cancer, obesity, malnutrition).

Recently, the European Working Group on Sarcopenia in Older People (EWGSOP2) published a revised consensus on the definition and diagnosis of sarcopenia, by focusing on low muscle strength, also called dynapenia, as a key characteristic of sarcopenia that is measured by hand grip strength (HGS) [8,11,12]. Indeed, the diagnosis of sarcopenia is confirmed by reduced appendicular skeletal muscle mass (ASM), in addition to dynapenia.

Different techniques can be used for the evaluation of muscle mass, such as dual X-ray absorptiometry (DXA) [13], computed tomography (CT) scanning [14] or (BIA) [15,16,17,18,19]. In the literature, sarcopenia is described as an unfavorable condition for patients with cancer disease because its presence is associated with more severe clinical conditions and longer hospital stays following surgery [20]. Additionally, it has been reported that the presence of sarcopenia can be used as a prognostic factor for mortality in both non-metastatic and metastatic patients with BC [21]. In a recent meta-analysis by Zhang et al. [22], BC patients, classified as sarcopenic had a greater mortality risk than non-sarcopenic BC patients.

Therefore, it is crucial to evaluate the presence of sarcopenia before starting cancer treatment in all patients, since it is strongly associated with chemotherapy-induced toxicity, post-operative complications and poor survival [23]. Indeed, low muscle mass at cancer diagnosis is associated with poor survival in patients with solid tumors [24].

To our knowledge, there are only a few studies in the literature that have specifically screened the presence of sarcopenia in Italian women with BC, according to the EWGSOP2. Thus, the objective of this study was to evaluate the prevalence of sarcopenia in a sample of Italian women with BC and a BMI lower than 30 kg/m^2^.

## 2. Patients and Methods

### 2.1. Patients

This is a cross-sectional study of baseline data collected from an ongoing randomized controlled trial in pre-and postmenopausal Italian women with stage 0-III BC, candidates for surgery and adjuvant/neoadjuvant therapy. Patients were consecutively recruited at the Oncology Unit, Department of Clinical Medicine and Surgery, Federico II University, Naples Italy, from September 2018 to June 2021.

Caucasian patients were consecutively included according to the following criteria: women, age ≥ 18 years, 18.5 ≤ BMI < 30 kg/m^2^ and available past medical history. Patients with metastatic cancer, BMI ≥ 30 kg/m^2^ and severe clinical conditions were excluded. In addition, a control group with similar age and BMI was selected from our database (Caucasian healthy women underwent a nutritional evaluation, aged between 35 and 75 years) with anthropometric and body composition data from the last two years.

The study was conducted in accordance with the Declaration of Helsinki, and it was approved by the local Ethics Committee of Federico II University (prot. n. 280/17). All patients gave their informed consent to participate in the study.

### 2.2. Anthropometry, Body Composition and Muscle Strength Assessment

Body weight and stature were measured according to standardized methods. Body weight was measured to the nearest 0.1 kg using a platform beam scale and stature to the nearest 0.5 cm using a stadiometer (Seca 709; Seca, Hamburg, Germany). Body weight and stature were used to calculate BMI (weight in kilograms divided by stature in meters squared).

Body composition was evaluated by the BIA method. BIA was performed using Human Im Plus II (DS Medica-Milan, Italy) at a room temperature of 22–25 °C in the 12 h fasted state after voiding the bladder and cleaning the surface of the skin to adhere the electrodes. Participants were asked to remain in the supine position for at least 10–15 min before starting the measurement, with lower limbs and upper limbs slightly abducted at 45° and 30°, respectively, to avoid any contact between the extremities and the trunk.

Resistance (R) and reactance (Xc) were measured at 50 kHz; PhA (degrees) = arctan (Xc/R) × (180/π) was calculated. Body composition assessment, namely, FFM and FM, was estimated using the Sun BIA equation [25]. ASM was calculated using the Sergi BIA equation [26].

Muscle strength was assessed by HGS, measured in both the dominant and nondominant hands with a Jamar dynamometer (JAMAR, Roylan, UK).

Patients performed the test standing with their upper limbs by their sides, and they were instructed to squeeze a dynamometer at maximal voluntary isometric contraction. The measurement was repeated three times alternately on both sides (dominant and nondominant hand), with 1 min apart to avoid fatigue, and the dominant hand was determined by asking subjects if they were right- or left-handed. The mean value was recorded in kilograms [27].

### 2.3. Definition of Sarcopenia

Sarcopenia was diagnosed according to the EWSGOP2 criteria. This new revised consensus suggests that the first parameter of sarcopenia is dynapenia evaluated with the HGS method (recorded in kilograms). The diagnosis of sarcopenia is confirmed by reduced ASM in addition to dynapenia and/or low physical performance. According to the cutoff points of Dodds RM et al. [28], dynapenia was diagnosed if HGS < 16 kg for females [8]. Low ASM was defined using the Studenski cutoffs: <15 kg for women [29]. Both conditions (dynapenia + low ASM) were considered for the diagnosis of sarcopenia. Instead, pre-sarcopenia was considered in the presence of either HGS value < 16 kg or ASM < 15 kg.

## 3. Statistical Analysis

The data obtained were analyzed by the SPSS (Version 27.0, IBM Corp, Armonk, NY, USA) software. Results are presented as the mean ± standard deviation (SD), and statistical significance was defined as *p* < 0.05. Differences between two groups (patients with BC and control group) were assessed by unpaired t-tests, while data were compared between four groups (sarcopenic BC patients, pre-sarcopenic BC patients, non-sarcopenic BC patients and control group) by using non-parametric analysis (Mann–Whitney test); the false discovery rate approach was performed to account for the multiple comparisons, and significant level (*p* < 0.05) was adjusted according to Benjamini and Hochberg [30]. HGS and ASM were adjusted for age, body weight and stature (UNIANOVA test). Categorical variables were compared by using the chi-square test.

## 4. Results

A total of 142 patients with BC participated in this study, but 20 were ruled out for the following reasons: 3 subjects did not meet the inclusion criteria and 17 left for personal reasons.

Thus, 122 patients, were included in this analysis. The anthropometric characteristics of patients with BC are summarized in Table 1. Patients with BC showing a mean age of 49.3 ± 11.0 years, an average body weight of 63.4 ± 7.4 kg. A total of 66 (54%) out of 122 patients had a BMI varying from 18.5 to 24.9 kg/m^2^, while 56 (46%) had a BMI range from 25 to 30 kg/m^2^. Additionally, as reported above, a control group with similar age and BMI to the patients was selected. In detail, healthy women aged between 35 and 75 years, with a body weight between 46 and 89 kg, of whom 45 (56.2%) were normal weight and 35 (43.8%) were overweight (Table 1).

A total of 45.9% of patients with BC were in the postmenopausal state, while 45.1% were pharmacologically induced. The histological characteristics of the tumor are reported in Table 2. Regarding cancer stage, 2.5% were stage 0, 41.2% were stage I, 39.5% were stage II, and 16.8% were stage III. A total of 69.6% of patients underwent quadrantectomy, and 30.4% underwent mastectomy. A total of 77% were estrogen receptor (ER) positive and 70.5% progesterone receptor (PR) positive, while 50.8% were human epidermal growth factor receptor 2 (HER2) positive. A total of 18.9% and 37.7% of the patients had started neoadjuvant and adjuvant chemo-radiotherapy, respectively, and 13.9% of women were in hormone therapy.

Muscle mass and muscle strength parameters, as well as PhA, are described in Table 3. Differences were observed between patients with BC and controls in terms of both quantitative (FM) and qualitative parameters (PhA). As such, patients with BC had lower values of PhA and HGS, and higher percentage of FM than controls.

Based on the EWGSOP2 guidelines, the prevalence of sarcopenia in the patients with BC was 13.9% (17/122), while none of the subjects in the control group had sarcopenia. In addition, among patients with BC, 31.1% (38/122) were pre-sarcopenic due to the presence of either HGS value < 16 kg or ASM < 15 kg. The remaining 54.9% (67/122) of patients were classified as non-sarcopenic, having both values in the normal ranges.

Data concerning anthropometry, body composition, PhA, muscle mass and strength between sarcopenic, pre-sarcopenic, non-sarcopenic BC patients and controls are shown in Table 4.

By comparing the results among the four groups, we found that sarcopenic patients with BC were older than non-sarcopenic and control group patients and had significantly lower body weight than in the other groups (Table 4).

Regarding body composition, differences were observed between sarcopenic BC patients compared to non-sarcopenic BC patients and control group in terms of both quantitative (FFM) and qualitative muscle parameters (PhA); in detail, the differences of PhA in sarcopenic BC patients compared to non- sarcopenic BC patients and control group were (−0.4 degrees, *p* = 0.045 and −0.5 degrees, *p* = 0.048), respectively.

Instead, the FM expressed in kilograms was significantly lower in sarcopenic BC patients than in non-sarcopenic BC (−4.4 kg, *p* = 0.024).

As in patients with sarcopenia, the group with pre-sarcopenia had both lower FFM and ASM than non-sarcopenic patients (−4.91 kg, *p* < 0.001) and control group (−3.47 kg, *p* < 0.001), respectively.

Additionally, after adjusting for age, body weight and stature, both HGS and ASM remain confirmed to be significantly lower in patients with sarcopenia than in other groups. Finally, menopausal state, tumor stage and therapy do not influence the presence and the grade of sarcopenia.

## 5. Discussion

The presence of sarcopenia in patients with cancer has been associated with reduced effectiveness of anti-neoplastic therapies, post-operative complications and poorer overall survival [23]. The poor outcomes may be related to higher drug toxicity rates in sarcopenic patients, which in turn may lead to dose reductions and delivering lower doses of effective cancer treatments. These phenomena could possibly be explained by the well-known association between lean and muscle mass and pharmacokinetics parameters, such as drug distribution, metabolism and the clearance of chemotherapeutic agents [24]. Due to the negative impact of sarcopenia on the outcome of cancer patients [24], we evaluated its prevalence in a population of women with early-stage BC cancer, by measuring body composition by BIA and muscle strength by HGS.

BIA is a portable, easy-to-use and inexpensive method for estimating fat mass (FM) and fat-free mass (FFM) in clinical settings [31,32,33,34]. The PhA, the most clinically relevant impedance parameter, is an index of cell membrane integrity and vitality, and it provides crucial information on cellular health and soft tissue hydration [35,36,37]. Low PhA values are generally associated with impaired muscle function, poor physical performance and low survival in different acute and chronic diseases, including cancer [36,37,38,39].

The compromised conditions of our patients with BC are also confirmed by the lower PhA values than in the controls.

Given its non-invasivity, BIA can be performed whenever necessary, without the limitations of other exams using radiation, such as DXA or CT scan, thus allowing a strict monitoring of the patient body composition. On the other hand, abdominal CT scan is not recommended for early stages of BC [40]. For this reason, the use of CT scan would not have allowed the inclusion of a large population of patients with BC [41].

The prevalence of sarcopenia in patients with BC was assessed according to the EWGSOP2 criteria. Sarcopenia was found in 13.9% of patients with BC and was absent in the control group. Moreover, a considerable number (31.1%) of patients with BC were pre-sarcopenic, showing similar characteristics to sarcopenic patients. These prevalence data are not irrelevant if we consider that the evaluated patients are relatively young, normal or overweight and with early-stage BC. The influencing factors, such as tumor stage, breast surgery, physical activity, may play a significant role [42].

Based on a recent systematic review [23], the prevalence of sarcopenia in patients with different types of cancer, evaluated before starting the treatment, was 38.6% of patients; unfortunately, only three studies [43,44,45] focused on a consensual definition of sarcopenia based on the EWGSOP published in 2010.

To our knowledge, the presence of sarcopenia according to the new EWGSOP2 guidelines in Italian patients with early-stage BC was evaluated only in one study by Bellieni et al. [46]. The study assessed sarcopenia in Italian older women with early-stage BC, obtaining a prevalence of 43%; the higher values found in that population can probably be explained by the older age of patients. In the literature, many studies analyzed the prevalence of sarcopenia in patients with cancer using different validated techniques for body composition analysis (DXA, CT or BIA) [8].

Previously, Oflazoglu et al. evaluated the prevalence of sarcopenia using the BIA method in a Turkish population with different types of cancers: breast, colorectal, pancreaticobiliary, genitourinary, lung, and head and neck [47]. The prevalence of sarcopenia was 16.7% in all patients (77/461), and it was 11.5% for those with BC (17/148). These results are similar to our data, but they considered the EWGSOP guidelines [10] and not the updated ones [8]. Similarly, the prevalence of sarcopenia using the previous criteria was evaluated in 98 survivors of BC (age > 60 years) among residents in Bogotà [48], showing a percentage of 22.4%. Another study conducted in a Brazilian population and using both BIA and HGS showed that the prevalence of sarcopenia was 18.6% in 60 oncologic patients and 10% in patients with BC using EWGSOP [49]. Recently, Ueno et al. [50] found a prevalence of sarcopenia of 12.2% in Asian patients with BC, by using the CT scan to measure skeletal muscle mass index. The prevalence of sarcopenia was similar to our study, although the method to assess body composition was different, and HGS was not evaluated.

Based on these controversial results and considering the widespread use of the previous guidelines for diagnosing sarcopenia, we analyzed our data according to the EWGSOP criteria as well [10]; sarcopenia was detected in 24 patients with BC, with a prevalence of 19.7%, which was higher than that observed by the EWGSOP2 definition [8]. Interestingly, this result was similar to the study conducted by Benavides-Rodríguez et al. [48], although the patients were older, but higher than Oflazoglu et al. [47] and Harter et al. [49].

Less is known about the body composition of women with early-stage BC, probably because the absolute recurrence and mortality rate in this population, to date, remains extremely low [51].

Considering the prevalence of sarcopenia observed in our patients, in our view, it is important to define the nutritional needs and physical activity programs to improve skeletal quality and strength (FFM, HGS and PhA). Body composition and the levels of physical activity should be assessed at baseline and periodically re-evaluated during the entire course of the treatment and over time, in the long-term follow-up, to suggest specific strategies (physical activity programs, aerobic exercise, strength training, resistance exercise therapy), which have been shown to reverse sarcopenia and were associated with improved quality of life [16,52,53,54].

Our study had some limitations. Firstly, this is a cross-sectional study, thus only baseline results are reported; follow-up data are ongoing and will be shortly available.

Secondly, although BIA is a valid and reliable tool for body composition analysis in clinical practice [16], it is not a gold-standard method; however, as already discussed, BIA allowed us to evaluate a larger group of patients with early-stage BC and to accurately follow up the possible changes of body composition during the therapeutic course, thereby guaranteeing the best chemotherapy dosing and tolerance.

Finally, patients BC with BMI ≥ 30 kg/m^2^ were excluded according to the study protocol, and the control group was retrospectively obtained from our dataset, and these may be considered further limitations.

## 6. Conclusions

In conclusion, we found that the overall proportion of sarcopenic and pre-sarcopenic patients in our population was somewhat higher than expected in a healthy population of women [41]. The use of BIA in association with HGS is relevant for identifying sarcopenia in BC patients, playing an important role for the patient’s outcome, in particular for early recognition, prompt intervention and periodical reassessment of the risk of sarcopenia and its associated complications.

## Figures and Tables

**Table 1 nutrients-14-01839-t001:** Characteristics of breast cancer (BC) patients and the control group.

		BC Patients (*n* = 122)			Control Group (*n* = 80)		
Age	years	49.4	±	11.0	48.2	±	10.0
Weight	kg	63.4	±	7.4	63.5	±	12.1
Stature	cm	161	±	7	161	±	6
BMI	kg/m^2^	24.6	±	3.0	24.5	±	4.1

Data are expressed as the mean ± SD. BMI = Body mass index. *p* values non-significant for all variables.

**Table 2 nutrients-14-01839-t002:** Clinical characteristics of patients with BC.

Tumor Stage	*n*	(%) *
0	3	2.5
I	49	41.2
II	47	39.5
III	18	16.8
Axillary lymph node metastasis		(%) **
Yes	48	41.7
No	67	58.3
Estrogen receptor status		(%) ^+^
Positive	91	77.1
Negative	27	22.9
Progesterone receptor status		(%) ^&^
Positive	79	70.5
Negative	33	29.5
Human epidermal growth factor receptor 2		(%) ^§^
Positive	60	50.8
Negative	58	49.2
Type of therapy		(%)
No therapy yet	36	29.5
Neoadjuvant chemotherapy	23	18.9
Adjuvant chemotherapy	46	37.7
Hormone therapy	17	13.9
Type of surgery		(%) ^$^
Quadrantectomy	80	69.6
Mastectomy	35	30.4
Menopausal status		(%)
Premenopausal	11	9.0
Postmenopausal	56	45.9
Induced menopause	55	45.1

Data unavailable for *n* (%):* 3 (2.4%); ** 7 (5.7%); ^+^ 4 (3.2%); ^&^ 10 (8.1%); ^§^ 4 (3.2%); ^$^ 7 (5.7%).

**Table 3 nutrients-14-01839-t003:** Body composition and hand grip strength measurements.

		BC Patients (*n* = 122)			Control Group (*n* = 80)		
FFM	kg	42.7	±	3.8	43.7	±	4.9
FM	kg	20.7	±	5.1	19.8	±	8.6
FM	%	32.3	±	5.1 *	29.9	±	8.2
ASM	kg	15.8	±	1.5	16.3	±	2.1
PhA	degrees	5.5	±	0.5 *	5.7	±	0.6
HGS	kg	19.2	±	5.6 *	21.0	±	4.2

Data are expressed as the mean ± SD. * *p* < 0.05 FFM = Fat-Free Mass; FM = Fat Mass; PhA = Phase Angle; HGS = Hand Grip Strength; ASM = Appendicular Skeletal Muscle Mass.

**Table 4 nutrients-14-01839-t004:** Individual characteristics and body composition of sarcopenic, pre-sarcopenic and non-sarcopenic patients and the control group.

			BC Patients		Control Group
		Sarcopenic *n* = 17	Pre-Sarcopenic *n* = 38	Non-Sarcopenic *n* = 67	*n* = 80
Age	years	55.8 ± 12.5 ^a^	51.4 ± 11.3 ^b^	46.6 ± 9.1	48.2 ± 10.0
Weight	kg	56.1 ± 4.8 ^c^	60.4 ± 5.5 ^b^	67.0 ± 6.8 ^d^	63.5 ± 12.1
Stature	cm	155 ± 7 ^a^	158 ± 6 ^a^	164 ± 6 ^d^	161 ± 6
BMI	kg/m^2^	23.4 ± 2.8	24.4 ± 3.1	25.0 ± 2.9	24.5 ± 4.1
FFM	kg	38.6 ± 1.9 ^c^	40.2 ± 2.5 ^a^	45.1 ± 2.9 ^d^	43.7 ± 4.9
FM	kg	17.5 ± 4.2 ^b^	20.2 ± 4.7	21.9 ± 5.2	19.8 ± 8.6
FM	%	30.8 ± 5.3	33.0 ± 5.2	32.3 ± 5.0	29.9 ± 8.2
ASM	kg	14.1 ± 0.8 ^c^	14.9 ± 0.9 ^a^	16.8 ± 1.2	16.3 ± 2.1
PhA	degrees	5.2 ± 0.5 ^a^	5.5 ± 0.5	5.6 ± 0.5	5.7 ± 0.6
HGS	kg	13.0 ± 2.0 ^c^	16.9 ± 4.1 ^a^	22.0 ± 4.2	21.0 ± 4.2

Data are expressed as the mean ± SD. Non-parametric test was used for statistical analysis ^a^
*p* < 0.05 vs. non-sarcopenic BC and control group; ^b^
*p* < 0.05 vs. non-sarcopenic BC; ^c^
*p* < 0.05 vs. pre-sarcopenic BC, non-sarcopenic BC and control group, ^d^
*p* < 0.05 vs. control group; FFM = Free-Fat Mass; FM = Fat Mass; PhA = Phase Angle; HGS = Hand Grip Strength; ASM = Appendicular Skeletal Muscle Mass.

## Data Availability

The data presented in this study are available on request from the corresponding author.

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
