# Peer review of "Prevalence of Sarcopenia in Women with Breast Cancer"

_nutrients, 2022, doi:10.3390/nu14091839_

Round 1

Reviewer 1 Report

This study examined the prevalence of sarcopenia in a sample of Italian women with breast cancer and then compared with a matched control group. Strength of this study is the use of low muscle strength (EWGSOP2) as a primary parameter of sarcopenia. It is considered a more reliable measure of muscle function and a better predictor of adverse outcomes.

My comments:

  1. Nutritional information is missing which would make this manuscript to fit better to the journal’s scope. The journal focuses on human nutrition and falls into category of "Nutrition & Dietetics".
  2. “This is a cross-sectional study of baseline data collected from an ongoing randomized controlled trial (prot. n. 280/17)...” I could not find this clinical trial to be registered in any clinical trial registry recognised by WHO and ICMJE.
  3. It is not clear the reason of using controls. Justification is needed.
  4. “…a control group, matched for age, body weight and BMI, was randomly selected from our database…” The authors matched cases and controls on body weight and BMI which is not a recommended approach as body weight is included in BMI. A more appropriate approach would be matching by age and BMI.
  5. In Table 4 the number of sarcopenic patients is low and non-parametric tests should be used. Furthermore, multiple testing corrections adjust p-values derived from multiple statistical tests are needed to correct for occurrence of false positives.
  6. Discussion needs improvements. For example, what are the implications of these findings for BC patients?
  7. “To our knowledge, this is the first time that the presence of sarcopenia was evaluated according to the new EWGSOP2 guidelines in Italian patients with early BC.” However, a previous study exists in Italian older women with early breast cancer admitted at the Breast Surgery Unit of the Fondazione Policlinico Universitario A. Gemelli IRCCS (https://doi.org/10.3390/jpm11040243).

Author Response

Dear reviewer,

Reviewer 2 Report

This is short straightforward analysis giving glimpse about sarcopenia among non-obese patients with early stage breast cancer before the treatment who also participated at a treatment clinical trial. Since this is expected to be a population with low degree of comorbidities, the prevalence of sarcopenia cca 14% seems relatively high. This is an interesting result.

Comments to the authors:

Better specify reasons why patients with BMI>30 were excluded. Some of these patients are at risk of sarcopenic obesity. Consider adding discussing limitations (BMI>30 excluded) in the discussion section.

 Were all patients female? Consider adding info to the manuscript.

 Major: Definition of sarcopenia needs clarifications. Lines 119-122: For the sentence: “The cutoff points of Dodds RM et al. [28] were used to identify dynapenia: males < 27 kg, females <16 kg [8].”  This is confusing. Are males required to weigh less than 27kg?  Or  what is XX <27kg  and what is XX<16 kg? To be sarcopenic, were patients required to have both dynapenia and low ASM ? Or were they required to have at least one of dynapenia or low ASM?

Clearly state the definition of sarcopenia used in this analysis.

Statistical methods: Line 127 consider reformulation: …ANOVA with Tukey’s post hoc test, as appropriate.

Statistical methods: Line 128: This reviewer is confused about the sentence: “HGS and ASM were adjusted for same individual parameters (UNIANOVA test).” Consider rewording and specifying. What are the same parameters? Was analysis of association of HGS and ASM with sarcopenia adjusted for “??? same parameters””?

Matching: Author states that BC patients were matched with controls by age, weight, BMI by the match randomly selected from the database. However there are 122 patients  and only 80 controls. Why do some patients not have the match?  Is there a difference between patients who were matched and who were not?

 Line 185 188. It is not surprising that HGS, ASM were lower in sarcopenic patients.  Both of these are part of the definition of sarcopenia as used for the analysis.  This seems like circular reasoning.  Consider justifying this analysis and clarifying the definition of sarcopenia 

Line 212 consider wording:  ….criteria, and then patients were compared …

Line 224 consider: … patients with early stage BC

Author Response

Dear reviewer,

Round 2

Reviewer 1 Report

Due to the link of nutrients to sarcopenia (PMID: 32545408) and its relatively high prevalence in this population it would be interesting to examine these association. It would be also interesting compare the nutritional status of BC patients with controls.

Prospective registration of clinical trials is an important safeguard against selective reporting and non-publication of research (http://www.icmje.org/recommendations/browse/publishing-and-editorial-issues/clinical-trial-registration.html).

Regarding the point 5, you should use the false discovery approach to account for multiple comparisons.

Author Response

Dear reviewer,

Reviewer 2 Report

Overall the authors answered the questions from the first review.

With adding a new text some grammar/formatting  glitches  (e.g line 87, 143) got in. Consider proofreading carefully the manuscript for grammar.

Instead of P=0.000 strongly consider P<0.001

Author Response

Dear reviewer,
